# DRSNFuse: Deep Residual Shrinkage Network for Infrared and Visible Image Fusion

**DOI:** 10.3390/s22145149

**Published:** 2022-07-08

**Authors:** Hongfeng Wang, Jianzhong Wang, Haonan Xu, Yong Sun, Zibo Yu

**Affiliations:** 1School of Mechatronical Engineering, Beijing Institute of Technology, Beijing 100081, China; 3120185177@bit.edu.cn (H.W.); 3120200259@bit.edu.cn (H.X.); 3120195181@bit.edu.cn (Y.S.); 3120200269@bit.edu.cn (Z.Y.); 2State Key Laboratory of Explosion Science and Technology, Beijing Institute of Technology, Beijing 100081, China

**Keywords:** image fusion, deep residual shrinkage network, channel-wise attention mechanism, auto encoder and decoder, artificial texture suppression

## Abstract

Infrared images are robust against illumination variation and disguises, containing the sharp edge contours of objects. Visible images are enriched with texture details. Infrared and visible image fusion seeks to obtain high-quality images, keeping the advantages of source images. This paper proposes an object-aware image fusion method based on a deep residual shrinkage network, termed as DRSNFuse. DRSNFuse exploits residual shrinkage blocks for image fusion and introduces a deeper network in infrared and visible image fusion tasks than existing methods based on fully convolutional networks. The deeper network can effectively extract semantic information, while the residual shrinkage blocks maintain the texture information throughout the whole network. The residual shrinkage blocks adapt a channel-wise attention mechanism to the fusion task, enabling feature map channels to focus on objects and backgrounds separately. A novel image fusion loss function is proposed to obtain better fusion performance and suppress artifacts. DRSNFuse trained with the proposed loss function can generate fused images with fewer artifacts and more original textures, which also satisfy the human visual system. Experiments show that our method has better fusion results than mainstream methods through quantitative comparison and obtains fused images with brighter targets, sharper edge contours, richer details, and fewer artifacts.

## 1. Introduction

Infrared images captured by infrared sensors to record the thermal radiations emitted by different objects are widely used in object detection and tracking [1,2]. They are robust to the influence of illumination variation and disguises such as objects in smoke. Infrared images provide distinct object boundaries, making it easier to locate targets. However, infrared images usually contain few high-frequency textures, which may be essential in target classification and tracking. In contrast, visible images provide rich texture information, whereas targets in visible images may not be easily observed due to the external environment, such as objects hidden in darkness. Multi-sensorial data fusion provides an efficient method in data analysis [3]. Infrared and visible image fusion (IVIF) can significantly preserve information from both infrared and visible images, satisfying the needs of the following CV tasks. Fused images can also provide perceptual scene descriptions for human eyes.

Recently, IVIF has received much attention, and various IVIF methods have been proposed in recent years [4,5]. IVIF algorithms can generally be divided into two groups: traditional methods and deep learning methods. The representative traditional methods include multi-scale transformation [6,7,8], sparse representation [9], subspace learning [10], and the saliency method [11]. Traditional methods usually exploit different feature extraction algorithms to overcome the variety of external environments. These methods are dedicated to improving a feature extraction and fusion strategy in manual ways for better fusion performance. The manually designed feature extraction and fusion strategy make the fusion methods more and more complex, increasing the time cost of image fusion. At the same time, traditional methods are less robust than deep learning methods due to the limitation of handcrafted feature extractors. At the same time, former experience plays an essential role in selecting the appropriate feature extraction and fusion strategy for different scenes, which also increases the cost of traditional methods.

Deep learning methods have achieved impressive accuracy and efficiency in IVIF. With the development of GPUs, IVIF methods based on deep learning can also perform real-time image fusion, even faster than traditional methods. Deep learning methods can be roughly divided into two groups: generative adversarial network (GAN)-based methods [12,13] and encoder–decoder-based methods [14,15,16].

Ma et al. proposed FusionGAN in [13] to perform image fusion. IVIF methods based on GANs usually include a generator and a discriminator. They play against each other to create images with realistic details. It is worth noticing that shortcomings also exist in deep learning methods. There is a significant difference between GANs performing IVIF and GANs with other tasks, such as single-image super-resolution [17,18,19]: no groundtruth exists in the IVIF task. The discriminator of FusionGAN takes visible images as groundtruth, comparing the fused images and original visible images. The discriminator of GANMcC [20] takes infrared and visible images as inputs at the same time and classifies inputs into infrared images, visible images, or fused images. However, GANMcC still cannot solve the absence of groundtruth, which is an obstacle to improving fusion performance in GAN-based methods.

In contrast to GAN-based methods, encoder–decoder-based methods usually take deep learning modules as encoders and decoders, extract features and then merge them into fused images. Li et al. exploited image multi-scale transformation in [14] to perform IVIF. However, Ref. [14] still took a handcrafted feature extraction module as the encoder. Therefore, some shortcomings similar to traditional methods also exist in the encoder of [14]. A similar situation also exists in [15], and the detail exposition will be discussed in Section 2.1. Li et al. proposed DenseFuse [16], which consists of two neural networks: an encoder and a decoder. The encoder extracts features into high domain maps, and the decoder reveals the high domain maps into fused images. Zhao et al. proposed an auto-encoder in DIDFuse [21], which extracts features into backgrounds and details. The purpose of IVIF is to keep the original information from both infrared images and visible images, which means that texture information cannot be ignored in IVIF. Hou et al. proposed VIF-Net in [22] performing unsupervised infrared and visible image fusion. Ma et al. proposed STDFusionNet in [23], which performs infrared and visible image fusion with target information. Although DIDFuse exploits a shallow network with four convolutional layers, some textures in fused images created by DIDFuse are covered with artificial textures. At the same time, DIDFuse pays the same attention to every channel in features, which means that objects in original images have the same weights as backgrounds. STDFusionNet takes extra segmentation information for image fusion in the training phase, whereas most aligned infrared and visible image pairs do not offer segmentation annotations.

This paper proposes a deep residual shrinkage network for infrared and visible image fusion (DRSNFuse). Faced with the lack of groundtruth, we propose an encoder–decoder-based method, which consists of an auto-encoder and an auto-decoder. The encoder extracts feature maps, and the decoder creates fused images from feature maps. We exploit residual shrinkage blocks to deepen the encoder while employing a global feature fusion module to fuse feature maps from different layers. On the one hand, a deeper network indicates a powerful feature extractor, which means more semantic information in feature maps and enables our method to be object-aware. On the other hand, residual shrinkage blocks and the global feature fusion module preserve more texture information from original infrared and visible images. Residual shrinkage blocks employed in our method introduce an attention mechanism into infrared and visible image fusion, which enables the network to pay weighted attention to objects and backgrounds.

Contributions. The contributions can be summarized as follows: (1) We proposed an end-to-end deep residual shrinkage network (DRSNFuse) for IVIF, which can perform object-aware image fusion. We employed residual shrinkage blocks and a global feature fusion module for encoding, which retains the original textures from infrared and visible images while extracting more semantic features. To the best of our knowledge, this is the first deep residual shrinkage network that processes three-domain images rather than one-domain signals, and our network has a state-of-the-art performance in IVIF. (2) We introduced an attention mechanism into the IVIF task, which guides our method to focus on objects in infrared and visible images. DRSNFuse separates objects and backgrounds apart, and objects in fused images generated by our method have a higher contrast with backgrounds. (3) We proposed an artificial texture loss to suppress the artifacts generated in decoding. Our artificial texture loss introduces a penalty for textures generated by fusion strategies, which imposes a limitation on recovering textures. (4) Experiments on RoadScene and TNO datasets reveal that our method can perform state-of-the-art infrared and visible image fusion compared with mainstream image fusion methods. DRSNFuse effectively fuses sharp edge contours from infrared images and texture information from visible images. The proposed method can suppress the artificial textures and preserve more original textures from infrared and visible images. Our method also achieved impressive performance in the quantitative evaluation.

## 2. Related Work

### 2.1. IVIF Methods Based on Deep Learning

In general, deep learning methods can be categorized into two groups: generative adversarial network-based methods and encoder–decoder-based methods.

Ma et al. proposed FusionGAN in [13], which consists of a generator and a discriminator. The generator takes infrared and visible images as input and output-fused images. The discriminator predicts the labels of images and enforces the generator to create images with more details from infrared and visible images. However, there are no original fused images serving as groundtruth in the IVIF task. Due to the absence of groundtruth, the discriminator of FusionGAN takes visible images as groundtruth. This strategy makes the fused images pay more attention to visible images, whereas the information included in infrared images may be ignored. Ma et al. enhanced the loss function in FusionGAN in [24] to keep edge information and details from original images. The optimized loss function attempts to retain more information from infrared images to reach a balance between infrared and visible images. Although the edge-enhancement loss function helps the generator focus on the edge information in infrared images, the discriminator still compares fused images with visible images. The detail loss function guides the generator to create images with more textures, which also leads to artificial textures in fused images. Ma et al. proposed GANMcC in [20], in which the discriminator takes infrared and visible images as groundtruth. The discriminator of GANMcC does not output true or false but predicts three labels: infrared images, visible images, or fused images. Infrared and visible images have equal importance in the discriminator of GANMcC, avoiding the fused images inheriting more information from one kind of original image without balance. However, the problem of groundtruth still cannot be ignored in GAN based method. Generally speaking, the absence of groundtruth is a vital obstruction for GAN-based methods in generating realistic fused images.

Encoder–decoder-based methods can also be divided into two sub-groups according to how encoders extract features: traditional encoder based methods and deep learning encoder based methods. Traditional encoder-based methods usually exploit handcrafted filters or optimization methods for image decomposition and then generate fused images with neural networks. Li et al. proposed a multi-scale transformation image fusion method in [14]. The encoder in [14] takes an optimization method to transform images from the spatial domain into base parts and detail content. After that, the decoder-based on VGG [25] merges features from base parts and detail content into the fused images. Lahoud et al. introduced an image fusion method exploiting filters in the decomposition stage [15]. In contrast to [14], Ref. [15] exploited a CNN module for further feature extraction. Furthermore, then, the feature maps from base parts and details are merged into the fused images in the fusion stage. In traditional encoder-based methods, the strategies of the image decomposition stage are still handcrafted, which also suffer from the detriments existing in traditional methods.

Similarly to traditional encoder-based methods, the deep learning encoder learns to extract features from original images, while the decoder manages to recover fused images from feature maps. Li et al. introduced an auto-encoder-based method DenseFuse in [16], a milestone of deep learning encoder-based methods. DenseFuse consists of two subnetworks: an encoder and a decoder. The encoder extracts features into high domain maps, and the decoder reveals the high domain maps into fused images. DenseFuse exploits a deep learning module as its encoder to replace the handcrafted filters or optimization methods. Zhao et al. proposed an auto-encoder in DIDFuse [21], which extracts features into backgrounds and details. The core idea of DIDFuse is to propose a CNN-based encoder, which can automatically transform images from the spatial domain to the background and detail domains. In brief, smooth areas in images are classified as backgrounds, while areas covered with textures are classified as details. Compared with DenseFuse, DIDFuse proposes an image decomposition loss to separate backgrounds and details from infrared and visible images. The decoder takes decomposed features as input and achieves better fusion performance than DenseFuse.

The advantages and open issues of mainstream deep learning methods are summarized in Table 1. In summary, deep learning encoder-based methods exploit convolutional neural networks to extract features from original infrared and visible images, and then merge feature maps into fused images with different strategies. Unlike object detection or tracking, in which semantic information from deep layers plays an important role, the purpose of IVIF is to merge texture information and edge information in fused images. Consequently, most deep learning-based extraction networks have shallow architectures to prevent texture information from vanishing during the broadcast through a deep network. Due to the encouragement of the detail loss function or adversarial loss function, the fusion strategies tend to create images with more textures, even though there may be no such textures either in original infrared images or visible images. These textures may lead to better IVIF results measured with specific metrics, but they may become drawbacks in the following tasks, such as object detection [26]. Moreover, they also provide wrong perceptual scene descriptions for human eyes, leading to wrong judgments.

### 2.2. Deep Residual Networks

He et al. [27] proposed deep residual networks (ResNets) for image recognition. Image recognition not only needs semantic information from the deeper layers of the networks but also evaluates the texture information from shallow layers. Unlike former convolutional neural networks, ResNets exploit shortcuts from shallow layers to deep layers, bringing texture features to the outputs. ResNets are widely used in one-stage object detection networks [28,29], and two-stage networks [30], in which ResNets perform as an excellent feature extractor. In ResNets, features are not only propagated layer by layer but also flow to the output through shortcuts. Generally speaking, the semantic information is extracted through the procession of backbone layers, while the texture information is maintained by means of shortcuts. ResNets can provide deeper backbone networks for feature extraction and, at the same time, maintain high-frequency information through shortcuts.

Zhao et al. proposed deep residual shrinkage networks in [31], which provides an efficient method for fault diagnosis. In contrast to ResNets, deep residual shrinkage networks add another sub-branch in residual blocks, generating thresholds to apply soft thresholding on the outputs of the backbone networks. Deep residual shrinkage networks are widely used in fault diagnosis [32,33,34], in which deep residual shrinkage networks recognize the target signal from noise. Yang et al. [35] introduced a fault diagnosis method of rotating machinery based on one-dimensional deep residual shrinkage networks, which significantly improves fault diagnosis accuracy.

In the mentioned methods, deep residual shrinkage networks are employed to process one-domain signals and achieve excellent performances. However, deep residual shrinkage networks have not been exploited in three-domain image tasks to our knowledge. Similarly to fault diagnosis, objects in original infrared and visible images also have more importance than backgrounds in the image fusion task. Therefore, our method exploits residual shrinkage blocks, guiding the network to focus on objects in original images.

## 3. Proposed Method

This paper proposes a deep residual shrinkage network for infrared and visible image fusion: DRSNFuse. Our method merges objective-aware transformation into the auto-encoder, with which DRSNFuse can separate the input images into base parts and details. Base parts represent areas that have similar pixel intensities in infrared and visible images, while details represent areas with different pixel intensities. Specifically, the objects contained in details are separated from backgrounds channel-wise. After that, we exploit an auto-decoder to generate images from base parts and details.

As shown in Figure 1, DRSNFuse consists of an auto-encoder and an auto-decoder, which takes infrared and visible images as inputs and performs end-to-end image fusion. The residual shrinkage blocks (RSBs) and global feature fusion (GFF) module in our encoder perform feature extraction on infrared and visible images, outputting infrared and visible features. The following base extractor and the detail extractor in our encoder separate the features into base parts and details. The base parts, details, and shallow features are then delivered to the decoder. The fusion layers in the decoder fuse base parts, details, and shallow features according to the fusion strategy, outputting fused features. After that, the decoding layers recover realistic images containing infrared and visible information from fused features. We enhance the loss function of DRSNFuse to suppress artificial textures, outputting images with few artifacts. The artificial texture loss helps DRSNFuse create realistic images and maintain original textures, which may be essential in following tasks such as object detection and tracking.

In the training phase, the encoder takes the infrared images and the visible images as inputs, outputting infrared base parts, infrared details, shallow infrared features, visible base parts, visible details, and visible shallow features. The decoder separately recovers the infrared images and the visible images from infrared data (infrared base parts, infrared details, infrared features) and visible data (visible base parts, visible details, visible features). In the inference phase, the encoder performs the same tasks as in the training phase. The decoder takes the infrared data and the visible data as inputs, merging base parts, details, and features from infrared and visible data separately. After that, the decoder generates fused images from the merged base parts, details, and features.

### 3.1. Network Architecture

The architecture of DRSNFuse is shown in Figure 2. There are four RSBs, one GFF module, and two convolutional layers working as base and detail extractors in the encoder. The architectural details of the encoder are shown in Table 2. Our encoder performs object-aware image decomposition, which separates objects and details into different channels in the feature map. The detail will be exhibited in Section 4.2.

First, DRSNFuse exploits four RSBs for feature extraction, and then the GFF module compresses the features from different layers into the infrared and visible feature maps. After that, the base extractor and the detail extractor separate the base parts and the details from feature maps. The base parts, the details, and features from RSB 1–2 are delivered to the decoder to recover fused images. In order to recover images with more realistic details, we introduced a GFF module to merge features from every RSB. At the same time, the encoder outputs feature maps from RSB 1–2, the output feature maps of which contain more texture information than the base parts and details generated by extractors.

The architecture of the mentioned RSBs is shown in Figure 3 and Table 3, which are similar to the block in [31]. We propose two types of residual shrinkage blocks: residual shrinkage block with channel-shared thresholds and residual shrinkage block with channel-wise thresholds. The backbone of RSBs consists of two convolutional layers, a soft thresholding layer and an activation layer. Additionally, there is a shortcut to bring features from the inputs to the deep layers and a thresholding branch to generate thresholds applied in the following soft thresholding. The network in [31] performs fault diagnosis on one-domain signals, which means it takes tensors in the shape of n×1 as inputs. However, in DRSNFuse, the encoder extracts features from original three-domain images, which takes tensors in the shape of n×H×W×C as inputs. RSBs can effectively preserve shallow features, which contain texture information from infrared and visible images. At the same time, due to RSBs avoiding the loss of texture information, deeper networks can be employed in IVIF tasks. Deeper networks perform better feature extraction and provide abundant semantic information for the following decoder to generate fused images [27]. A residual shrinkage block with channel-wise thresholds also introduces an attention mechanism into the IVIF task, which helps our method pay more attention to the object channels.

The decoder consists of four convolutional layers, which merge base parts, details, and features, recovering fused images. The architectural details of the decoder are shown in Table 4. The convolutional layer group 4 concatenates base parts and details from infrared and visible images, outputting feature maps with base and detail information. The convolutional layer groups 5 and 6 separately merge shallow features from RSB 2 and RSB 1 into the fused feature maps. The following convolutional layer group 7 employs reflection padding to generate more realistic edge areas and recover fused images from feature maps. We exploit a sigmoid layer in convolutional layer group 7 as the activation function, limiting the pixel intensities of fused images into interval (0,1).

Unlike the former auto-encoder networks such as DIDFuse, we adapt a deep residual shrinkage network for feature extraction rather than fully convolutional networks. In IVIF tasks, shallow features can hardly resist the broadcast of deep, fully convolutional networks. Consequently, fully convolutional network-based encoders usually employ shallow architectures, such as the four-layer encoder in DIDFuse [21]. On the one hand, the residual shrinkage network maintains texture information throughout the network with minor damage. On the other hand, the deeper network can extract more semantic information than shallow, fully convolutional networks [27,36]. Our method performs an object-aware image decomposition, separating objects and backgrounds into different channels. Our method also exploits an attention mechanism in the image fusion task, which guarantees objects in fused images a higher contrast with backgrounds.

### 3.2. Loss Function

We incorporate the loss functions from some state-of-the-art IVIF approaches [15,16,21] and propose our loss function that satisfies the needs of IVIF, suppressing artificial textures.

#### 3.2.1. Encoder Loss

In the fusion of infrared and visible images, the areas with similar pixel intensities in infrared and visible images can be merged with little transformation, while the areas with different pixel intensities require an effective fusion strategy. We take the areas with similar pixel intensities as base parts and areas with different pixel intensities as details to perform image fusion more meticulously. The purpose of the encoder loss is to enforce the encoder to separate base parts and details apart for the decoder to recover fused images. Our encoder loss consists of a base part loss and a detail loss. The base part loss encourages the base extractor to focus on similar areas in infrared and visible images. On the contrary, the detail loss guides the detail extractor to extract different features from infrared and visible images.

Base part loss.Base parts are defined as areas with similar pixel intensities in infrared and visible images, so the infrared base parts should enjoy high similarity to the visible base parts. The base part loss is computed as (Equation 1). We exploit a tanh function to limit the base part loss to the interval of (−1,1), which can avoid gradient exploding problems in training. As shown in (Equation 1), minor base part loss indicates a smaller gap between base part features from infrared and visible images, which coincides with the definition of base parts.
(1)$$L_{\mathrm{base}}=\tanh\left(\left|B^{inf}-B^{vis}\right|\right)$$
(2)$$B^{inf}=B\left(E\left(I^{inf}\right)\right),\;B^{vis}=B\left(E\left(I^{vis}\right)\right)$$

Here, $B^{inf}$ and $B^{vis}$ denote infrared and visible base parts as shown in (Equation 2). $I^{inf}$ and $I^{vis}$ denote infrared images and visible images. E(*) denotes the residual shrinkage blocks and the global feature fusion module. B(*) denotes the base extractor.

Detail loss. Details in infrared and visible images are the critical areas in image fusion. From the definition of details, we can conclude that the gap between infrared detail maps and visible detail maps should be distinct, so we exploit (Equation 3) to compute the detail loss. We also exploit a tanh function to limit the detail loss into interval (−1,1), which ensures a balance in the training phase between the base extractor and the detail extractor. As shown in (Equation 3), minor detail loss means a more distinct gap between the detail maps from the infrared and visible images. After that, the encoder loss is computed as (Equation 5).
(3)$$L_{\mathrm{detail}}=-\tanh\left(\left|D^{inf}-D^{vis}\right|\right)$$
(4)$$D^{inf}=D\left(E\left(I^{inf}\right)\right),\;D^{vis}=D\left(E\left(I^{vis}\right)\right)$$
(5)$$L_{\mathrm{encoder}\;}=\alpha_{1}\cdot L_{\mathrm{base}\;}+\alpha_{2}\cdot L_{\mathrm{detail}\;}$$

Here, $D^{inf}$ and $D^{vis}$ denote the infrared and visible details, respectively, as shown in (Equation 4). D(*) denotes the detail extractor. $\alpha_{1}$ and $\alpha_{2}$ denote trade-off weights.

#### 3.2.2. Decoder Loss

In the training phase, the decoder learns to recover original images from base parts, details, and shallow features. The decoder loss consists of pixel-wise loss, structural similarity loss, gradient loss, and artificial texture loss.

Pixel-wise loss. A natural and straightforward way is to enforce the decoder’s output to be the original images by minimizing the pixel-wise loss, which has been proven effective in some state-of-the-art approaches [24]. The pixel-wise loss guides the decoder to recover images with similar pixel intensities as the original images. The pixel-wise loss is computed as (Equation 6). Minor pixel-wise loss means a higher pixel intensity similarity between the recovered and original images.
(6)$$L_{\mathrm{pixel}-\mathrm{wise}}={\left(\bar{I^{inf}}-I^{inf}\right)}^{2}+{\left(\bar{I^{vis}}-I^{vis}\right)}^{2}$$
(7)$$\begin{array}{cc} \; & \bar{I^{inf}}=R\left(B^{inf},D^{inf},F^{inf}\right),\\ \; & \bar{I^{vis}}=R\left(B^{vis},D^{vis},F^{vis}\right) \end{array}$$

Here, $\bar{I^{inf}}$ and $\bar{I^{vis}}$ denote infrared and visible images recovered by the decoder as shown in (Equation 7). $F^{inf}$ and $F^{vis}$ denote shallow features from infrared and visible images. $R\left(*,*,*\right)$ denote images recovered by the decoder.

Structural similarity loss. Wang et al. [37] propose a method of measuring structural similarity (SSIM) that compares local patterns of pixel intensities between two images. The structural similarity loss is computed as (Equation 8). Minor structural similarity loss means a higher structural similarity between the recovered and original images.
(8)$$L_{\mathrm{SSIM}}=2-\mathrm{SSIM}\left(\bar{I^{inf}},I^{inf}\right)-\mathrm{SSIM}\left(\bar{I^{vis}},I^{vis}\right)$$

Here, SSIM(*,*) denotes the SSIM calculator.

Gradient loss. Solutions to MSE optimization problems often lack high-frequency content, which results in images covered with overly smooth textures. Therefore, we adopt gradient loss to guarantee texture agreement between recovered and original images. Since visible images are enriched textures, our gradient loss focuses on visible images, forcing the decoder to pay more attention to recovering textures from visible images. The gradient loss is computed as (Equation 9).
(9)$$L_{\mathrm{gradient}}=\left|\nabla \bar{I^{vis}}-\nabla I^{vis}\right|$$

Here, ∇ denotes the gradient operator.

Artificial texture loss. The former IVIF methods focus on adding more textures in fused images without limitation, which may improve results such as spatial frequency. However, some textures do not exist either in original infrared or visible images: artificial textures. Artificial textures can significantly improve some IVIF results, such as the average gradient, whereas they also cover original textures contained in infrared and visible images, which is against the purpose of IVIF. The gradient loss increases high-frequency content in fused images, making them sharper. However, the gradient loss without suppression tends to introduce artificial textures into images. These artificial textures significantly reduce the gradient loss, but they also obscure the original textures of images, which may be fatal for following tasks. Artificial texture loss is proposed to suppress the artificial textures encouraged by gradient loss. The artificial texture loss also focuses on visible images, similarly to the gradient loss. The artificial texture loss is computed as (Equation 10). After that, the decoder loss is computed as (Equation 11).
(10)$$\begin{array}{cc} L_{\mathrm{artificial}}= & \sum_{x=1}^{W-1}{\left({\bar{I^{vis}}}_{x+1,*}-{\bar{I^{vis}}}_{x,*}\right)}^{2}+\\ \; & \sum_{y=1}^{H-1}{\left({\bar{I^{vis}}}_{*,y+1}-{\bar{I^{vis}}}_{*,y}\right)}^{2} \end{array}$$
(11)$$L_{\mathrm{decoder}}=\beta_{1}\cdot L_{\mathrm{pixel}-\mathrm{wise}}+\beta_{2}\cdot L_{\mathrm{SSIM}}+\beta_{3}\cdot L_{\mathrm{gradient}}+\beta_{4}\cdot L_{\mathrm{artificial}}$$

Here, ${\bar{I^{vis}}}_{x,*}$ and ${\bar{I^{vis}}}_{*,y}$ denote the x−th row and the y−th column of the recovered images, respectively. *W* and *H* denote the width and height of the recovered images, respectively. $\beta_{1}$, $\beta_{2}$, $\beta_{3}$ and $\beta_{4}$ denote the trade-off weights.

## 4. Experiments and Evaluations

### 4.1. Training Details

We compare our method with other state-of-the-art IVIF methods, including ADF [7], HMSD-GF [8], VSMWLS [38], FusionGAN [13], DenseFuse [16], GANMcC [20], and DIDFuse [21]. We implement our model with PyTorch, and all the following experiments are performed on a single NVIDIA RTX 3090 GPU (NVIDIA Corporation, 2788 San Tomas Expressway, Santa Clara, CA 95051, US.) Table 5 shows our system requirements.

Datasets and preprocessing. We experimentally validated our proposed method on the RoadScene dataset [39] and the TNO dataset [40]. The RoadScene dataset consists of 221 aligned infrared and visible image pairs containing rich scenes such as roads, vehicles, and pedestrians. These images in RoadScene are highly representative scenes from the FLIR video. TNO Image Fusion Dataset contains multispectral nighttime imagery of different scenarios captured with different multiband camera systems, widely used in the evaluation of IVIF tasks. Table 6 shows details of the datasets used in the following experiments. We randomly divided RoadScene into the training set (181 pairs), the validation set (20 pairs), and the test set (20 pairs) according to the classic proportion of 8:1:1. We also randomly selected 20 image pairs from TNO as the test set, containing several representative scenes such as smoke, men, and trees. All images are in grayscale. In the training phase, we applied random cropping on the original images, which transforms the images into the shape of 128×128.

Training of DRSNFuse. In the encoder, we set the trade-off weights $\alpha_{1}=1$, $\alpha_{2}=0.5$. Furthermore, in the decoder, we set the trade-off weights $\beta_{1}=1$, $\beta_{2}=2.5$, $\beta_{3}=10$, $\beta_{4}=2\times 10^{-8}$. The trade-off weights $\alpha_{1}$ and $\alpha_{2}$ control the base part loss and the detail loss within a similar scale in the training phase of the encoder. Furthermore, the trade-off weights $\beta_{1}$, $\beta_{2}$, $\beta_{3}$ have the same impact in the training phase of the decoder as $\alpha_{1}$ and $\alpha_{2}$. The trade-off weight $\beta_{4}$ controls the suppression performance of the artificial texture loss, avoiding over-smooth fusion images. Our network is trained from scratch. The weights in each convolutional layer are initialized with a zero-mean Gaussian distribution with a standard deviation of 0.02, while the biases are initialized with 0. The weights in the fully connected layers branch are initialized with a zero-mean Gaussian distribution with a standard deviation of 0.1, while the biases are initialized with 0. We adopted Adam optimizers for the encoder and the decoder. The learning rates for the optimizers are initially set to $1\times 10^{-3}$ and then reduced to 95% after every epoch. There are 0.13 million trainable parameters in our channel-wise encoder and 0.10 million trainable parameters in our channel-wise decoder. As for the channel-shared DRSNFuse, there are 0.12 million trainable parameters in our encoder and 0.10 million trainable parameters in our decoder. The training is terminated after 150 epochs, and the states of the network are recorded.

### 4.2. Image Decomposition

Our method performs image decomposition with its encoder, and the deposed features are shown in Figure 4. The first and the second rows show infrared images and visible images. The third and the fourth rows exhibit base parts from infrared and visible images. The fifth and the sixth rows contain details from infrared and visible images.

As mentioned above, base parts represent areas that are similar in infrared and visible images. Compared with details, the differences between infrared base parts and visible base parts are relatively small, as shown in the third and fourth rows. The fifth and sixth rows contain details from infrared images and visible images, and we exhibit two representative channels from detail feature maps for every image. As shown in the fifth row, objects from infrared images are separated from the backgrounds. Specifically, the upper images in the fifth row contain original backgrounds from infrared images, and the lower images focus on objects and contain few backgrounds. In contrast with former methods, our method adapts residual shrinkage blocks for the backbone, which applies learnable soft thresholding on features. On the one hand, residual shrinkage blocks preserve shallow features throughout the whole net. On the other hand, the learnable soft thresholding guides channels of features to focus on different areas, which plays a similar role to the attention mechanism. Consequently, different areas are separated into various channels. Our encoder also applies a similar decomposition to visible images, as shown in the sixth row. Unlike infrared images, there is no distinct pixel intensity difference between the objects and backgrounds in greyscale visible images. Furthermore, there are no extra target annotations in IVIF datasets. Objects in visible images can hardly be separated from backgrounds as accurate as in infrared images.

### 4.3. Subjective Performance Evaluation

Figure 5 exhibits several representative image pairs. The first row shows original infrared images, and the original visible images are shown in the second row. The following rows contain fused images generated by different methods, including traditional methods (ADF, HMSD-GF, VSMWLS) and deep learning methods (FusionGAN, DenseFuse, GANMcC, DIDFuse, our method). The regions in small red boxes are shown in the bigger boxes, which reveal the performance gap between different methods.

In the first column, we can hardly recognize the letters in fused images generated by some methods. Especially in HMSD-GF, although the fused image generated by HMSD-GF seems lighter than others, they also lose information contained in the dark areas. The road sign in fused images generated by our method has better contrast than other methods, which provides a more precise edge contour, making letters easier to recognize. In the second column, the fused images generated by ADF and DenseFuse can only provide a dim shadow of the target, which means the target pixel intensity is significantly lower than other methods. Although FusionGAN and GANMcC perform better than ADF and DenseFuse, we still cannot obtain a clear outline of the target. The fused target generated by DIDFuse is wrapped with artificial textures, which may be fatal in the following tasks, such as object detection. Compared to other methods, our method provides a brighter fused target with a more precise edge contour and fewer artificial textures. In the third and fourth columns, the fused image generated by our method also has more clear details than other methods, which means our method may perform better in the following tasks, such as object tracking. In the third column, we can obtain clearer and richer details in the fused image generated by our method. In the fourth column, the fused image generated by our method has a better contour and details than other methods. As shown in the fifth column, the fused images generated by our method seem a little darker than HMSD-GF. Although the background of the running man in our method is darker than in other methods, pixels from the running man have a higher intensity. Lighter images cannot promise better fusion performance, but higher contrast makes objects clear. The reason is that the soft thresholding residual shrinkage blocks apply learnable suppression on the feature maps. At the same time, we exploit a sigmoid layer in our network, as shown in Figure 2, which limits the pixel intensities of fused images to the interval (0, 1), and enhances images with higher contrast, as shown in Figure 6. The upper images are original infrared and visible images. Taking the original images as inputs, the sigmoid layer in our network outputs the images with higher contrast, as shown in the second row.

Generally speaking, our method can generate fused images with higher contrast, brighter targets, sharper edge contours, richer details, and fewer artifacts than other methods.

### 4.4. Objective Performance Evaluation

We exploit eight metrics to evaluate the quality of fused images: mutual information (MI), average gradient (AG), entropy (EN), standard deviation (SD), spatial frequency (SF), sum of correlations of differences (SCD), structural similarity (SSIM), and visual information fidelity (VIF). As shown in Table 6, we randomly choose image pairs as the test set from RoadScene and TNO. Several image pairs in the test set are shown in Figure 7. The upper images in every row are infrared images, and the lower images are paired visible images. As shown in Figure 7, the test set contains many different scenes under varying illumination, such as streets, buildings, and forests. The objective comparison results on the test sets are shown in Table 7. Furthermore, the test results of each image pair are exhibited in Figure 8 and Figure 9.

Table 7 indicates that our method has the best MI, AG, EN, SD, SF, SCD, and the second best VIF on RoadScene. The largest MI suggests that our fused images retain more information from original infrared and visible images. The largest AG shows that our method performs sharper image fusion than other methods. The largest EN demonstrates that fused images generated by our method contain more abundant information than others. The largest SD indicates that our method can generate fused images with the best contrast. The largest SF demonstrates that our fused images can provide richer edges and textures. The largest SCD suggests that fused images generated by our method enjoy a higher similarity with original infrared and visible images. The second VIF indicates that our fused images also satisfy the human visual system. The SSIM of our method slightly falls behind the traditional methods (HMSD-GF and VSMWLS) but also has the best performance in deep learning methods. As for the test on TNO, our method has the AG, EN, SF, SCD, and the second best MI, SD, and VIF. The SSIM of our method on TNO is behind the traditional methods (HMSD-GF and VSMWLS) but also has the second best performance in deep learning methods. As shown in Table 7, our method has a better performance on RoadScene than TNO. Because we only exploit 181 image pairs from RoadScene as our training set, without any image pairs from TNO. RoadScene contains 221 image pairs, while there are only 42 in TNO. Although both RoadScene and TNO consist of infrared and visible image pairs, differences still exist in RoadScene and TNO, such as scene and illumination. At the same time, considering the size of TNO, we only exploited 181 image pairs from RoadScene but no image pair from TNO, which leads to the relatively low performance on TNO.

In summary, DRSNFuse can generate sharper images with more high-quality information. At the same time, fused images created by DRSNFuse also contain richer details and satisfy the human visual system.

The average inference time of different methods is provided in Table 8. The traditional methods (i.e., ADF, HMSD-GF, and VSMWLS) are tested on a desktop with a 5.10 GHz Intel Core i7 CPU. The deep learning methods (i.e., FusionGAN, DenseFuse, GANMcC, DIDFuse, and our method) are also tested on the mentioned desktop with an NVIDIA Geforce RTX 3090. Our method performs the second best image fusion speed. Even though DenseFuse has a shorter inference time compared with our method, the fusion result of DenseFuse is far from ours. At the same time, our method can satisfy the need for real-time infrared and visible image fusion.

## 5. Conclusions

In this paper, we proposed DRSNFuse, a novel object-aware infrared and visible image fusion method based on a deep residual shrinkage network. We exploited residual shrinkage blocks in our network, which introduces an attention mechanism in the IVIF task. Our method can separate objects and backgrounds into different channels, improving the image fusion performance. At the same time, the deep residual shrinkage network maintains a shallow texture information throughout the whole network, which also extracts more semantic information with a deeper backbone compared with other deep learning methods based on fully convolutional networks. We introduced artificial texture loss into the auto-decoder, with which the decoder can create fused images with fewer artifacts. The artificial texture loss helped the decoder balance original features and fusion textures. The experimental results showed that, compared with some state-of-the-art methods, DRSNFuse has a better performance according to the qualitative and quantitative comparison. DRSNFuse also satisfies the human visual system which generates fused images which contain higher contrast, brighter targets, sharper edge contours, and richer details.

In future work, we will further investigate the segmentation of objects and backgrounds. Precise segmentation and elaborate fusion strategy separately adapted for objects and backgrounds may achieve better performance, especially for the following tasks, such as object detection and tracking. We will also focus on dataset processing. From the gap between the results on the test sets from RoadScene and TNO, we can conclude that the variety of training sets plays an essential role in the performance of deep learning methods. Unlike object detection tasks with large-scale datasets such as COCO and VOC, the limitation of datasets is the main obstacle in the IVIF tasks. In order to broaden the IVIF datasets, we will also investigate the methods of infrared images and visible image alignment.

## Figures and Tables

**Figure 1 sensors-22-05149-f001:**
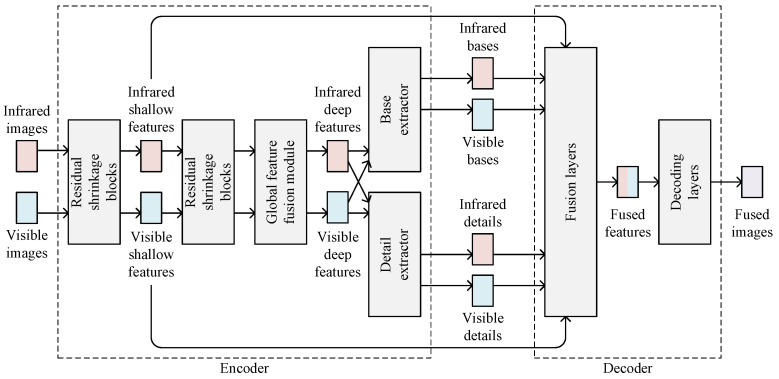
Frame of DRSNFuse.

**Figure 2 sensors-22-05149-f002:**
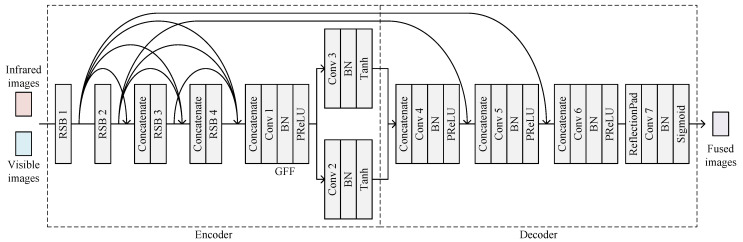
Architecture of DRSNFuse.

**Figure 3 sensors-22-05149-f003:**
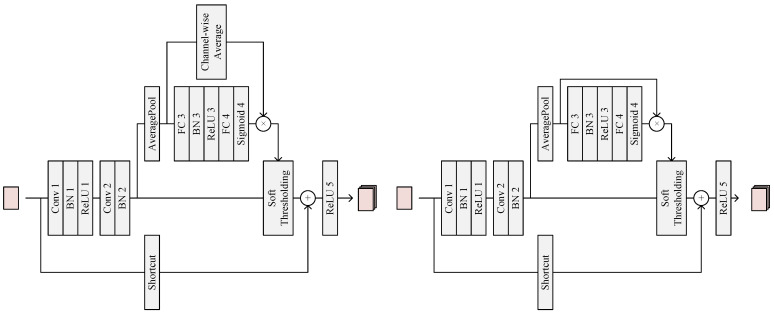
Residualshrinkage block. The left figure shows a residual shrinkage block with channel-shared thresholds. The right figure shows a residual shrinkage block with channel-wise thresholds.

**Figure 4 sensors-22-05149-f004:**
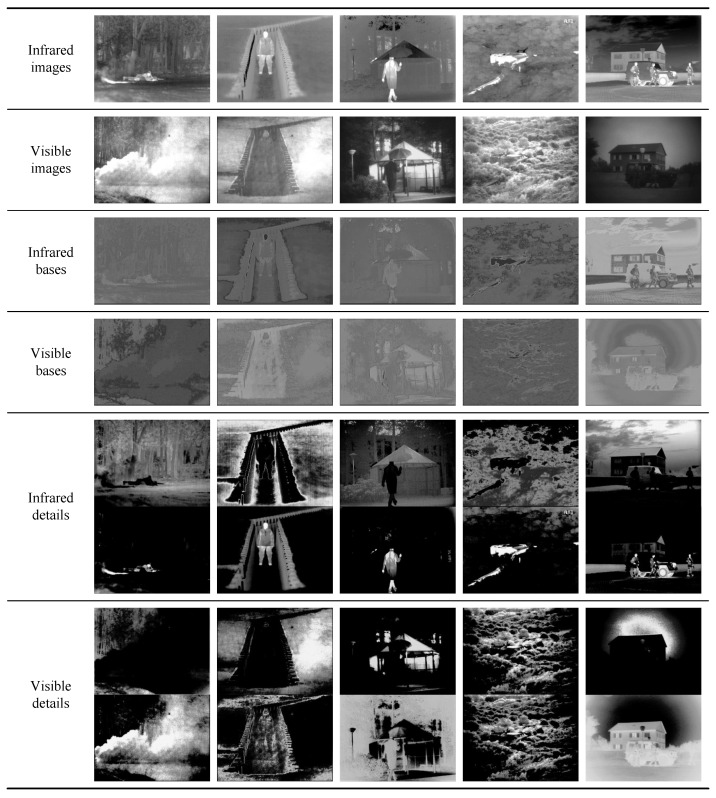
Base parts and details from infrared images and visible images.

**Figure 5 sensors-22-05149-f005:**
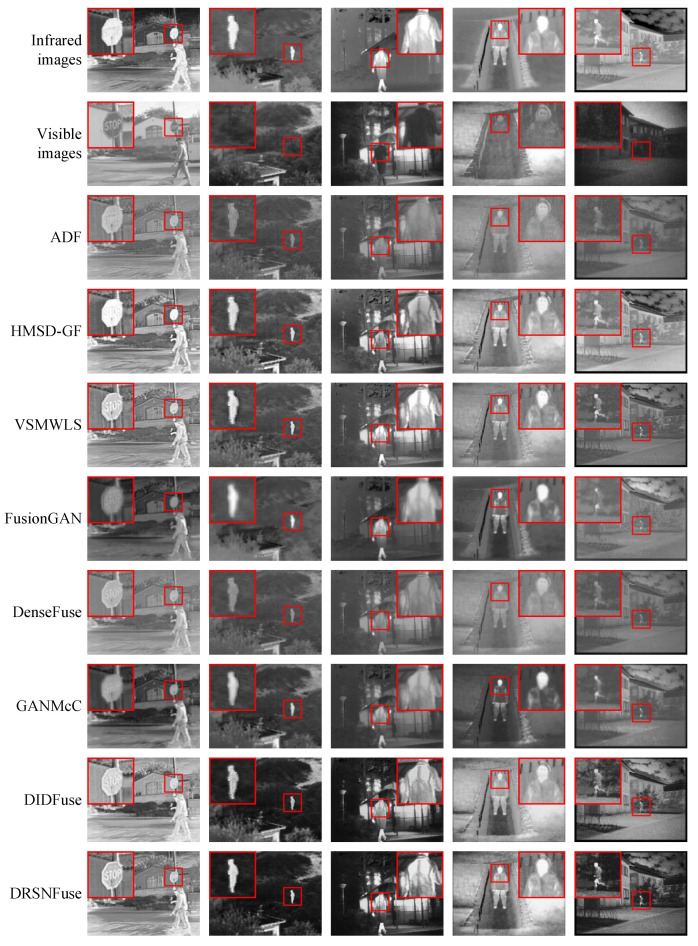
Qualitative results for different methods.

**Figure 6 sensors-22-05149-f006:**
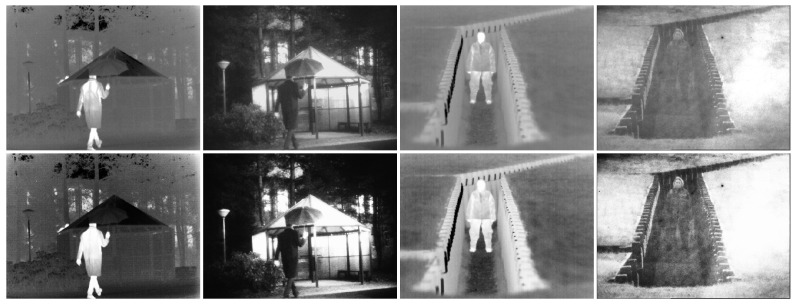
Sigmoid layer enhances images with higher contrast.

**Figure 7 sensors-22-05149-f007:**
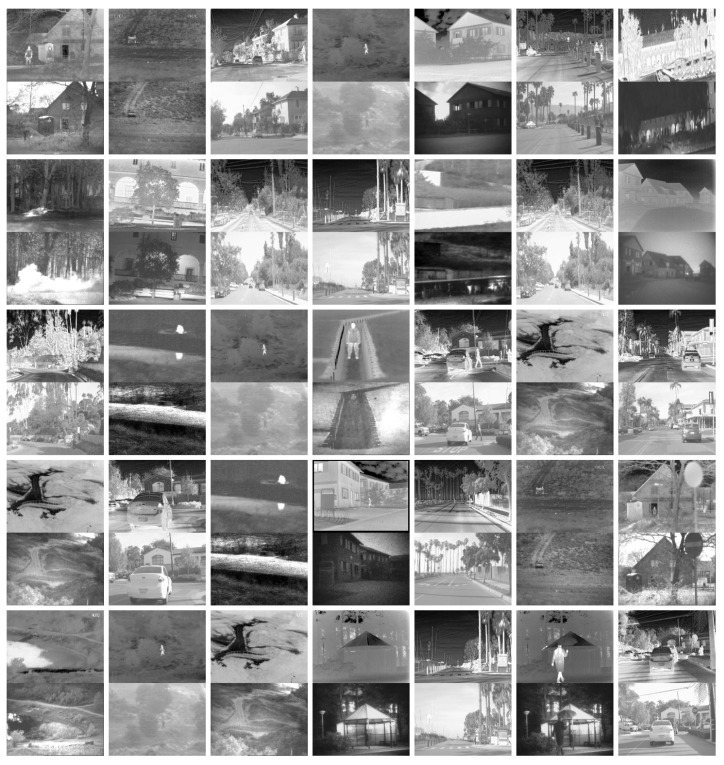
Infrared and visible image pairs in the test set.

**Figure 8 sensors-22-05149-f008:**
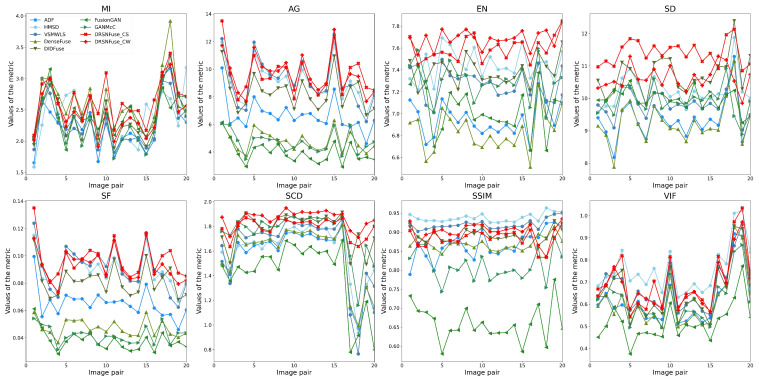
Fusion results of pairs in RoadScene.

**Figure 9 sensors-22-05149-f009:**
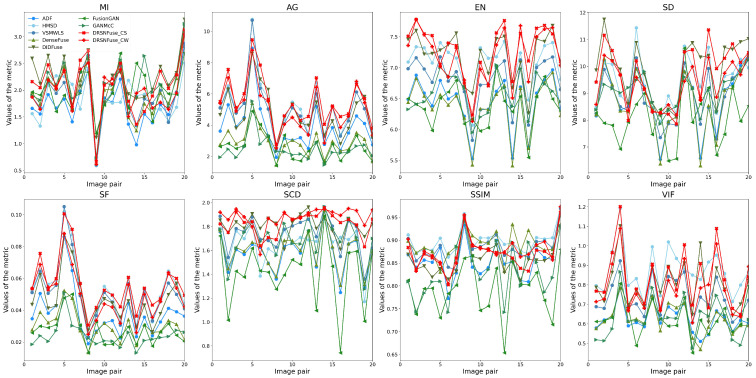
Fusion results of pairs in TNO.

**Table 1 sensors-22-05149-t001:** Mainstream deep learning methods summary.

Groups		Representative Methods	Advantages	Challenges
GAN-based methods		FusionGAN, GANMcC	Unsupervised image fusion with GANs	Absence of groundtruth
Encoder–decoder-based methods	Traditional encoders	Refs. [14,15]	Learnable decoders	Handcrafted feature extraction
	Deep learning encoders	DenseFuse, DIDFuse, VIF-Net	Learnable encoders and decoders	Relatively shallow networks

**Table 2 sensors-22-05149-t002:** Architecture of the auto-encoder in DRSNFuse.

Block	Chl_in	Kernel Num.	Kernel Size	Stride	Padding
RSB 1	1	16	3	1	1
RSB 2	16	16	3	1	1
RSB 3	32	16	3	1	1
RSB 4	48	16	3	1	1
Conv 1	64	16	1	1	0
Conv 2	16	64	3	1	1
Conv 3	16	64	3	1	1

**Table 3 sensors-22-05149-t003:** Architecture of the residual shrinkage block.

Layer	Kernel Num.	Kernel Size	Stride	Padding	Chl_in	Chl_out
ResidualShrinkage Block with Channel-Shared Thresholds						
Conv 1	Chs	3	1	1	-	-
Conv 2	Chs	3	1	1	-	-
FC 3	-	-	-	-	Chs	Chs
FC 4	-	-	-	-	Chs	1
Residual Shrinkage Block with Channel-Wise Thresholds						
Conv 1	Chs	3	1	1	-	-
Conv 2	Chs	3	1	1	-	-
FC 3	-	-	-	-	Chs	Chs
FC 4	-	-	-	-	Chs	Chs

**Table 4 sensors-22-05149-t004:** Architecture of the auto-decoder in DRSNFuse.

Layer	Kernel Num.	Kernel Size	Stride	Padding
Conv 4	64	3	1	1
Conv 5	32	3	1	1
Conv 6	16	3	1	1
Conv 7	1	3	1	0

**Table 5 sensors-22-05149-t005:** System requirements.

CPU	Intel 10700K
GPU	NVIDIA RTX 3090
OS	Ubuntu 20.04
Language	Python 3.8 with PyTorch 1.11.0

**Table 6 sensors-22-05149-t006:** Datasets used in experiments.

	Dataset (Pairs)	Illumination	Average Size
Training	RoadScene—train (181)	Daylight&Nightlight	514×302
Validation	RoadScene—validation (20)	Daylight&Nightlight	514×302
Test	RoadScene—test (20)	Daylight&Nightlight	514×302
	TNO(20)	Daylight&Nightlight	597×450

**Table 7 sensors-22-05149-t007:** Results of quantitative evaluation on TNO and RoadScene. The bold values are the best results. And the underline values rank second.

Method	ADF	HMSD-GF	VSMWLS	FusionGAN	DenseFuse	GANMcC	DIDFuse	DRSNFuse_CS	DRSNFuse_CW
RoadScene									
MI	2.252	2.410	2.274	2.614	2.412	2.407	2.317	**2.636**	2.458
AG	6.668	9.044	9.190	4.800	8.324	3.977	4.720	9.604	**9.630**
EN	6.948	7.447	7.282	6.841	7.386	7.034	7.297	7.595	**7.682**
SD	9.349	10.217	9.844	9.242	10.732	9.975	9.925	**11.400**	10.624
SF	0.065	0.089	0.091	0.047	0.082	0.038	0.042	**0.097**	0.093
SCD	1.556	1.612	1.624	1.579	1.780	1.416	1.752	1.793	**1.863**
SSIM	0.866	**0.938**	0.920	0.869	0.892	0.659	0.811	0.884	0.901
VIF	0.619	**0.758**	0.678	0.611	0.657	0.517	0.618	0.704	0.687
TNO									
MI	1.712	1.825	1.899	1.954	**2.194**	1.995	2.098	2.101	1.977
AG	3.881	5.229	4.989	2.888	4.774	2.614	2.493	**5.508**	5.011
EN	6.432	7.075	6.800	6.371	7.187	6.370	6.576	**7.283**	7.180
SD	8.754	9.448	8.874	8.702	**9.896**	8.080	9.048	9.694	9.338
SF	0.038	0.053	0.050	0.030	0.047	0.027	0.024	**0.056**	0.049
SCD	1.579	1.658	1.739	1.598	1.807	1.421	1.696	1.824	**1.877**
SSIM	0.856	**0.898**	0.885	0.891	0.859	0.791	0.832	0.871	0.879
VIF	0.624	**0.870**	0.738	0.623	0.799	0.629	0.627	0.824	0.788

**Table 8 sensors-22-05149-t008:** Average inference time. The bold values are the best results. And the underline values rank second.

Method	ADF	HMSD-GF	VSMWLS	FusionGAN	DenseFuse	GANMcC	DIDFuse	DRSNFuse_CS	DRSNFuse_CW
RoadScene									
Inference time	0.114s	0.164s	0.301s	0.212s	**0.007s**	0.456s	0.017s	0.015s	0.014s
TNO									
Inference time	0.219s	0.337s	0.603s	0.396s	**0.012s**	0.851s	0.099s	0.028s	0.027s

## Data Availability

Not applicable.

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
