# Peer review of "DRSNFuse: Deep Residual Shrinkage Network for Infrared and Visible Image Fusion"

_sensors, 2022, doi:10.3390/s22145149_

Round 1
Reviewer 1 Report
Paper Summary:
This paper addresses the problem of Infrared and visible image fusion (IVIF). The authors propose an object-aware image fusion method based on a deep residual shrinkage network (DRSNFuse). DRSNFuse exploits residual shrinkage blocks for image fusion and introduces a deeper network in IVIF tasks. Besides, a novel image fusion loss function is proposed to obtain better fusion performance and suppress artifacts. Experiments show that the proposed method has better fusion metrics than mainstream methods through quantitative comparison and obtains fused images with brighter targets, sharper edge contours, richer details, and fewer artifacts.
Paper Strengths:
This paper proposes an object-aware image fusion method a novel image fusion loss function to obtain better fusion performance and suppress artifacts. Experiments show that the proposed method has better fusion metrics than mainstream methods through quantitative comparison and obtains fused images with brighter targets, sharper edge contours, richer details, and fewer artifacts. The proposed method can suppress the artificial textures and preserve more original textures from infrared and visible images.
Paper Weaknesses:
1、The visible image should be a three-channel color image, why the visible image in this paper is a grayscale image?
2、The advantages of the methods proposed by the authors are obvious, and I would like to know the number of parameters and computation of these algorithms.
3、The trade-off weight of β4 = 2 × 10−8 is too small. Why is this? Can you do an experiment with different weights?
Reviewer 2 Report
The paper presents a Deep Residual Shrinkage Network for Infrared and Visible Image Fusion. The authors claim to present a novel methodology on the topic by proposing an encoder-decoder architecture with residual networks. In general, the article has been well written on an important subject with a solid set of experiments. However, the article needs major revisions to be accepted for publication.
First of all, abbreviations should be avoided in the abstract and a thorough revision in terms of the proper usage of the English language is needed.
In Section 2, the authors are kindly requested to provide a summary table with the advantages and the open issues of each method to inform in a more concrete way about the advantages and the open challenges on the subject field.
The authors may want to consider adding the following related references, among others:
1. J. Ma, L. Tang, M. Xu, H. Zhang and G. Xiao, "STDFusionNet: An Infrared and Visible Image Fusion Network Based on Salient Target Detection," in IEEE Transactions on Instrumentation and Measurement, vol. 70, pp. 1-13, 2021.
2.Kotsiopoulos, Thanasis, et al. "Deep multi-sensorial data analysis for production monitoring in hard metal industry." The International Journal of Advanced Manufacturing Technology 115.3 (2021): 823-836.
3. R. Hou et al., "VIF-Net: An Unsupervised Framework for Infrared and Visible Image Fusion," in IEEE Transactions on Computational Imaging, vol. 6, pp. 640-651, 2020, doi: 10.1109/TCI.2020.2965304.
Please provide citations about these sentences:
-
‘’’…At the same time, due to RSBs avoiding the loss of texture information, deeper networks can be employed in IVIF tasks. Deeper networks perform better feature extraction and provide abundant semantic information for the following decoder to generate fused images…’’’
-
‘’’…On the one hand, the residual shrinkage network maintains texture information throughout the network with minor damage. On the other hand, the deeper network can extract more semantic information than fully convolutional networks. …”
The authors should provide the corresponding references in Section 3.1:
-
“We incorporate the loss functions from some state-of-the-art IVIF approaches and propose our loss function that satisfies the needs of IVIF, suppressing artificial textures.”
“In the encoder, we set the trade-off weights α1 = 1, α2 = 0.5. 329
And in the decoder, we set the trade-off weights β1 = 1, β2 = 2.5, β3 = 10, β4 = 2 × 10−8.“ - The authors should provide an explanation regarding the loss’s hyperparameters setup. How they are chosen and why.
“The average inference time of different methods is provided in Table 7. The traditional 434 methods (i.e., ADF, HMSD-GF, and VSMWLS) are tested on a desktop with a 5.10 GHz Intel Core i7 CPU. The deep-learning methods (i.e., FusionGAN, DenseFuse, GANMcC, DIDFuse, and our method) are also tested on the mentioned desktop with an NVIDIA Geforce RTX 3090. Our method performs the second image fusion speed. Even though DenseFuse has a shorter inference time compared with our method, the fusion result of DenseFuse is far from ours. At the same time, our method can satisfy the need for real-time infrared and visible image fusion.” – Is the inference execution time of the Deep Neural networks tested without the use of the GPU? If not, the authors should provide also the execution time without the use of the GPU.
Round 2
Reviewer 1 Report
I have read the response. However, I still feel the current version exists many English grammar mistakes. For example (I just check the abstract),
a practical approach to obtaining-> to obtain
which also satisfies -> which also satisfy
fusion metrics -> fusion results
The authors need an editing service or letter from a colleague who has assisted with editing.
Author Response
Thank you very much for your comments.
We revised our paper carefully, especially on English grammar. And we corrected several mistakes similar to the ones mentioned.
Best regards.
Reviewer 2 Report
My comments have been addressed. I would recommend though a careful proofread of the manuscript.
Author Response
Thank you very much for your comments.
We proofread our paper carefully according to your comments.
Best regards.